# Fecal Microbiota Transplant in Two Ulcerative Colitis Pediatric Cases: Gut Microbiota and Clinical Course Correlations

**DOI:** 10.3390/microorganisms8101486

**Published:** 2020-09-27

**Authors:** Andrea Quagliariello, Federica Del Chierico, Sofia Reddel, Alessandra Russo, Andrea Onetti Muda, Patrizia D’Argenio, Giulia Angelino, Erminia Francesca Romeo, Luigi Dall’Oglio, Paola De Angelis, Lorenza Putignani

**Affiliations:** 1Area of Genetics and Rare Diseases, Unit of Human Microbiome, Bambino Gesù Children’s Hospital, IRCCS, 00165 Rome, Italy; quagliariello.andrea@gmail.com (A.Q.); sofia.reddel@opbg.net (S.R.); 2Department of Laboratories, Unit of Parasitology, Bambino Gesù Children’s Hospital, IRCCS, 00165 Rome, Italy; alessandra.russo@opbg.net; 3Department of Laboratories, Bambino Gesù Children’s Hospital, IRCCS, 00165 Rome, Italy; andrea.onettimuda@opbg.net; 4Academic Department of Pediatrics, Unit of Immune and Infectious Diseases, Bambino Gesù Children’s Hospital, IRCCS, 00165 Rome, Italy; patrizia.dargenio@opbg.net; 5Digestive Surgery and Endoscopy Unit, Bambino Gesù Children’s Hospital IRCCS, 00165 Rome, Italy; giulia.angelino@opbg.net (G.A.); efrancesca.romeo@opbg.net (E.F.R.); luigi.dalloglio@opbg.net (L.D.); paola.deangelis@opbg.net (P.D.A.); 6Department of Laboratories, Unit of Parasitology and Area of Genetics and Rare Diseases, Unit of Human Microbiome, Bambino Gesù Children’s Hospital, IRCCS, 00165 Rome, Italy; lorenza.putigani@opbg.net

**Keywords:** gut microbiota, fecal microbiota transplantation (FMT), Pediatric Ulcerative Colitis (UC), inflammatory bowel disease (IBD), gut bacterial ecology

## Abstract

Fecal microbiota transplantation (FMT) is a promising strategy in the management of inflammatory bowel disease (IBD). The clinical effects of this practice are still largely unknown and unpredictable. In this study, two children affected by mild and moderate ulcerative colitis (UC), were pre- and post-FMT monitored for clinical conditions and gut bacterial ecology. Microbiota profiling relied on receipts’ time-point profiles, donors and control cohorts’ baseline descriptions. After FMT, the improvement of clinical conditions was recorded for both patients. After 12 months, the mild UC patient was in clinical remission, while the moderate UC patient, after 12 weeks, had a clinical worsening. Ecological analyses highlighted an increase in microbiota richness and phylogenetic distance after FMT. This increase was mainly due to *Collinsella aerofaciens* and *Eubacterium biforme*, inherited by respective donors. Moreover, a decrease of *Proteus* and *Blautia producta*, and the increment of *Parabacteroides*, *Mogibacteriaceae*, *Bacteroides eggerthi*, *Bacteroides plebeius*, *Ruminococcus bromii*, and *B*
*Bacteroides*
*ovatus* were associated with remission of the patient’s condition. FMT results in a long-term response in mild UC, while in the moderate form there is probably need for multiple FMT administrations. FMT leads to a decrease in potential pathogens and an increase in microorganisms correlated to remission status.

## 1. Introduction

Inflammatory bowel disease (IBD) is a chronic inflammatory condition characterized by aberrant immune activation in susceptible individuals that leads to relapsing inflammation of the gastrointestinal (GI) tract [1]. Despite the numerous therapies developed for IBD, a large proportion of patients does not respond to the available drugs [2]. Furthermore, some concerns have emerged about their safety and potential long-term consequences [3]. It is known that gut microbiota in IBD patients is characterized by a condition of dysbiosis [4], thus its tracking has been proposed as a new diagnostic and prognostic tool to interfere with the disease’s natural history [5]. Hence, the imbalance in microbiota composition provides the rationale for therapeutic manipulation of the gut microbiota driven by nutritional interventions, probiotic administration and fecal microbiota transplantation (FMT). However, studies on prebiotics and probiotics have provided ambiguous results on the amelioration of IBD symptoms [6]. Thus, researchers have tried to develop new interventions to modulate or to reset gut microbiota composition; among others, FMT is one of the most promising.

FMT consists of an administration of fecal matter collected from a selected “healthy” donor to a recipient through colonoscopy, oral or other administration routes [7]. The success of FMT for the treatment of *Clostridium difficile* infection has been proven [8,9]. However, several articles have already reported on FMT’s potential therapeutic use in GI and GI-related diseases, including IBD [10,11,12,13,14,15,16,17].

The literature on pediatric FMT in IBD remains limited, often focused on the patient’s response, and lacking in a description of gut microbiota ecology pre- and post-FMT [18,19,20,21,22,23,24,25,26,27,28].

Herein, we report preliminary results from the first Italian study regarding FMT in children affected by IBD, focusing on microbial ecology and clinical patterns in the pre- and post-FMT course.

## 2. Materials and Methods

### 2.1. Subject Recruitment, Sample Collection and FMT Procedures

The first pilot study in Italy of FMT for Pediatric Inflammatory Bowel Diseases (PIBD) was developed at the Bambino Gesù Children’s Hospital (OPBG) in Rome: protocol n. 1107_OPBG_2016. The study was designed and conducted in accordance with relevant guidelines and regulations, as well as ethical principles for medical research involving human subjects (WMA Declaration of Helsinki). Each subject of the pilot study was evaluated by a multidisciplinary team, belonging to the FMT Transplantation Committee of the OPBG, composed by the Digestive Surgery and Endoscopy Unit (selection of patients, visits and exams, endoscopy evaluation, FMT infusion procedures), the Parasitology and Human Microbiome Units (screening procedures for donor selection and gut microbiota analyses for donors and recipients), and the Immunology and Infectious Disease Unit (infectious surveillance before/after FMT).

Patients’ inclusion criteria were: diagnosis of IBD, age 6–18 years, mild-to-moderate disease (10 < Pediatric Ulcerative Colitis Activity Index [PUCAI] < 65, 12.5 < weighted Pediatric Crohn’s Disease Activity Index [wPCDAI] < 57.5), relapsing course with poor control by traditional treatments, informed consent released. Patients’ exclusion criteria were: adult subjects; subjects with active GI infections; subjects that had changed medical therapy within last 4 weeks; subjects with a central venous catheter in place; subjects who were critically ill or who had comorbid medical illness.

During the period from May to October 2017, the first two patients affected by ulcerative colitis (UC) were enrolled and herein discussed. PUCAI [29] was assessed by direct interview with patients at the Digestive Surgery and Endoscopy Unit. Calprotectin values were estimated through CALPRTEST kit (EUROSPITAL, Italy) following manufacturer procedures. Values lower than 50 μg/gr were negative, in the range 50–100 μg/g weakly positive and higher than 100 μg/g positive. Further information about patient treatment and disease severity as well as a summary of the patient data are included in Table 1.

Donors were selected among first-degree relatives, aged 18–60 years, with general healthy conditions (both physical than psychological), excluding subjects with chronic, infectious or onco-hematological disease, as well as subjects with recent exposure to select drugs [30]. Screening procedures were optimized for pediatric FMT according to OPBG Protocol for FMT, ed. 11-09-2017, including deep evaluation of donor’s opportunistic pathogens/commensals presence and microbiota profiling. The donor stools were collected at home and delivered to OPBG as soon as possible and within 6 h since evacuation. Stools were weighted and mixed with sterile saline buffer at ratio 50/200_gr/mL_. The suspension was homogenized with a Stomacher 400 Circulator (Seward, UK) and filtered under biological safety cabinet. The final fecal suspension was then immediately delivered to the Digestive Surgery and Endoscopy Unit for the infusion. The OPBG FMT procedure consisted of endoscopic instillation of fecal preparation from fresh stool, after recipient conventional colon lavage by polyethylene glycol. The preparation was released in the cecum (colonic disease) or in duodenum-jejunum (small bowel disease). Follow-up visits were scheduled after 4 (T_1_), 8 (T_2_) and 12 (T_3_) weeks, including microbiota analyses. Endoscopic follow-up was planned after 12 weeks (T_3_). A microbiota profile matching was evaluated for each donor/recipient couple, employing three stool replicates, and providing a diagnostic report of microbiota maps.

For the analysis of microbiota, donors’ and recipients’ microbiota maps were compared during the entire FMT time-course with the maps of a group of age-matched healthy subjects (10 adults and 10 adolescents). The healthy controls were enrolled at the Human Microbiome Unit at the OPBG (Ethic Committee Protocol No. 768.12), based on the following criteria: absence of any chronic diseases, absence of gastro-intestinal infections, no antibiotic or pre/probiotic therapies in the previous 2 months and omnivorous diets.

### 2.2. SrRNA Metagenomic Sequencing

Bacterial DNA extraction was performed using QIAmp Fast DNA Stool Mini Kit (Qiagen, Hilden, Germany) according to the manufacturer’s instructions. Then, DNA of the V3-V4 regions from 16S rRNA was amplified using primer pairs 16S_F 5′-(TCG TCG GCA GCG TCA GAT GTG TAT AAG AGA CAG CCT ACG GGN GGC WGC AG)-3′ and 16S_R 5′-(GTC TCG TGG GCT CGG AGA TGT GTA TAA GAG ACA GGA CTA CHV GGG TAT CTA ATC C)-3′, according to MiSeq rRNA Amplicon Sequencing protocol (Illumina, San Diego, CA, USA). PCR reaction was performed following the instructions provided by Illumina’s protocol using a 2x KAPA Hifi HotStart ready Mix. Amplicons were cleaned-up using AMPure XP beads (Beckman Coulter Inc., Beverly, MA, USA), and a second amplification step was performed to obtain a unique combination of barcoded Illumina Nextera forward and reverse adaptor-primers. Library purification was performed using AMPure XP beads and then quantified using the Quant-iT™ PicoGreen^®^ dsDNA Assay Kit (Thermo Fisher Scientific, Waltham, MA, USA). Finally, the library was diluted in equimolar concentrations (4 nM). Samples were sequenced through the Illumina MiSeq DX platform, according to the manufacturer’s specifications, to generate paired-end reads of 300 base-length.

Reads were analyzed for their quality, length and chimera presence using the Qiime v. 1.8 pipeline [31]. Sequences were organized into Operational Taxonomic Units (OTUs) with 97% clustering threshold of pairwise identity using the VSEARCH-based consensus classifier [32]. OTUs’ were aligned against the Greengenes 13_08 database using the PyNAST v. 0.1. tool [33] with a 97% similarity for bacterial sequences.

### 2.3. Statistical Analyses

Aligned sequences were used to build a phylogenetic tree [34]. Statistical analyses were computed using R packages *phyloseq* for α and β diversity [35]; the *adonis* function in the R package *vegan* was used to perform the PERMANOVA test of β-diversity of patients’, donors’ and the control group’s samples with 9999 permutations. Differences in taxa relative abundance were assessed using the *DESeq2* package, by Wald test. Multiple testing corrections of the significant OTUs used the Benjamini-Hochberg method for false discovery rate [36].

Sequencing reads and the associated metadata are available at NCBI Bioproject: PRJNA517389.

## 3. Results

### 3.1. Specific Lines of FMT Protocols’ Management

Patients and donors were selected according to the inclusion criteria of the protocol. All clinical details are shown in Table 1.

### 3.2. Clinical Course

Two UC patients were discussed in the herein study. They both had a relapsing disease, with recurrent episodes of bloody diarrhea and abdominal pain, and had received multiple cycles of steroids/antibiotics during the year preceding FMT. After each family’s screening, their fathers were selected as stool donors. FMT was performed by colonoscopy in both cases.

Patient 1 was in maintenance therapy with mesalazine. Endoscopic assessment at the time of FMT (T_0_) revealed a left sided colitis (Mayo score: 0 in the ascending-transverse-descending colon, 2 in the sigmoid-rectum) (Figure 1, Panels A and B).

FMT follow-up was uneventful, and patient reported the absence of symptoms. Colonoscopy after 12 weeks (T_3_) revealed improvement of disease activity (Mayo score: 0 in the ascending-transverse-descending colon, 1 in the sigmoid-rectum) (Figure 1, Panels C and D). After 12 months, the patient was in clinical remission (Table 1).

Patient 2 was in maintenance therapy with mesalazine and azathioprine. Endoscopic assessment at the time of FMT (T_0_) revealed a pancolitis (Mayo score: 1 in the ascending-transverse colon, 2 in the descending colon-sigmoid-rectum) (Figure 1, Panels E and F). No complication occurred after FMT, and the patient reported clinical improvement with a reduction of bowel movements. After 16 weeks (T_3_), a relapse with bloody diarrhea occurred, treated with a course of antibiotic therapy with metronidazole. Follow up colonoscopy (T_3_) was similar to the baseline exam (Mayo score: 1 in the ascending-transverse colon, 2 in the descending colon-sigmoid-rectum) (Figure 1, Panels G and H). Even though the metronidazole administration at 16 weeks post-FMT can be classified as adverse event linked to patient condition severity, the collection of the following sample (T_4_) was performed only to describe the complete evolution of microbiota ecology during the clinical course. The patient was further evaluated after 16 weeks (T_4_) for a booster of FMT, but clinical relapse occurred again, and anti-TNF treatment was started (Table 1).

Calprotectin values seemed to not be influenced by FTM for P1, while a reduction of about half pre-FMT values was observed for P2. PUCAI values showed a clear post-FMT improvement for both patients (Table 1).

### 3.3. Gut Microbiota Ecology

UniFrac analysis revealed that donors’ samples were perfectly included within the control’s (CTRL) cluster and that all recipients’ samples were clustered together and separated from healthy CTRLs and donors (Appendix A).

The same analysis, for patient 1 (P1) and patient 2 (P2), showed not only a clear differentiation between recipient and donor samples, but also between samples at time of FMT (T_0_) and all other post-FMT samples (T_1_, 4 weeks; T_2_, 8 weeks; T_3_, 12 weeks; T_4_, 16 weeks) (Appendix A). The Shannon index demonstrated an increment of microbiota richness after FMT (starting from T1) compared to the diversity at T_0_ for both P1 and P2 (Appendix A). For P_2_, this increment reached its peak at T_2_ then decreased at later time points (Appendix A). The microbiota profile for each time point was evaluated at phylum level (Appendix A).

Comparing the microbiota composition of patients and respective donors, it was observed that two OTUs (*Collinsella aerofaciens* and *Eubacterium biforme*) were totally absent at T_0_ while present in donors and post-FMT samples, thus these taxa were derived by donors trough FMT (Figure 2, Panels A and B).

These acquired OTUs constituted a high percentage of the post-FMT microbiota for both patients. In samples collected from patient P1, we observed an increment, from 8% (T_1_) to 11% (T_2_), of these two species during the two months after FMT (Figure 2, Panel A). Whereas, in samples from P2, the percentage of donor species increased more slowly: they rose to 4% after the first month (T_1_), reached their peak (15%) after 3 months and then decreased almost to the T_1_ value (6%) at T_4_ (Figure 2, Panel B).

The microbial ecology of P1 and P2 was directly compared to describe which microbiota traits were shared between the two patients. Hierarchical cluster analysis, based on the Jaccard distance, revealed the existence of two main clusters: group 1 cluster was composed of T_1_, T_3_ and T_4_ samples from P2, and group 2 cluster was composed by all time-points of P1 and T2 from P2. The T_0_ sample from P2 was placed on a separate branch as outgroup (Appendix A). Comparing the composition of groups 1 and 2, it seemed that the cluster subdivision was correlated, above all, to the significant presence of *Prevotella copri*, *Lactobacillus* spp. and *Enterococcus* spp. in the group 1 cluster, and to Lachnospiraceae, *Sutterella* and *Bacteroides fragilis* presence in the group 2 cluster (Appendix A).

### 3.4. Microbial Ecology and Clinical Course

Interestingly, the clusters’ frame, based on the bacterial ecological structure, did not reflect their clinical condition as described above in paragraph “Clinical Course”. Therefore, to explore which bacterial species were most associated with the clinical status, samples were divided into two groups based on their respective clinical features, remission (RE) and active disease (AD) (Figure 3, Panel A), and compared to microbiota profiles of donors and controls. Three bacterial markers were found to be associated with the AD condition: *Proteus*, *Blautia producta* and *Ruminococcus gnavus*. On the other hand, *Parabacteroides*, Mogibacteriaceae, *Bacteroides eggerthi*, *Bacteroides plebeius*, *Ruminococcus bromii*, and *Bacteroides ovatus* were incremented in RE group (Figure 3, Panel B).

## 4. Discussion

The role of FMT in the treatment of pediatric UC is still debated, although the safety and tolerability of this clinical practice have already been assessed [16,17,28]. Recent reviews examined the current state of knowledge on FMT in IBD, and concluded that translational research on FMT is important in order to transfer its potential to real clinical management [15,37]. Our findings demonstrate in our patients an improvement of clinical conditions closely related to FMT. The FMT influenced the gut microbiota structure, decreasing the abundance of some bacterial species mainly associated with the moderate symptoms of the disease (such as *Proteus* or *B. producta*) and leading to the acquisition of species such as *C. aerofaciens* and *E. biforme,* which probably exert beneficial effects. However, other species (*e.g*., *P. copri*, *Enterococcus* spp. and *Lactobacillus* spp.) remained within the patients’ microbiota community, thus probably concurring to clinical UC relapse.

From a clinical point of view, our results demonstrate that for mild UC, a single FMT acted on disease remission, while it seemed not to prevent relapses in the moderate form. In fact, for P1, the FMT procedure positively influenced PUCAI and Mayo scores, with a decrease of their values after FMT, but without any effect on calprotectin values, consistent with a previous report [27]. For P2, PUCAI and calprotectin decreased, while Mayo scores remained unchanged.

Beta diversity analysis of the entire set of samples revealed two important findings. First, donor samples perfectly fitted within the variability spectrum of the CTRL cohort samples, thus indicating that, based on microbiota profiling, the donors were reliable candidates for FMT. Secondly, recipients’ samples, both before and after FMT, were characterized by a different microbial composition than the CTRL.

P1 post-FMT samples seemed to be somewhat closer to the CTRL group than to the P2 post-FMT samples; this finding correlates well with the amelioration of the clinical status of P1. When comparing the microbial ecology of recipients against their respective donors (without the CTRL cohort), a third interesting finding was observed: all post-FMT samples were phylogenetically distant not only from their respective donor but also from their own T_0_ samples. This indicates that FMT profoundly changed their gut microbial structure and led to a new ecology that was distant from both their T_0_ and from the donors’ ecology. This evidence was also confirmed by Shannon analysis that showed how microbial richness and biodiversity increased in both recipients after FMT, as also reported by Goyal and co-workers [28]. Particularly, P1 reached higher diversity values than the donor, while P2’s diversity increased slightly, perhaps due to the severity of P2’s initial clinical condition.

Of note, the relative abundance of two species, *C. aerofaciens* and *E. biforme*, which were completely absent at T_0_, greatly increased after FMT in both recipients. *C. aerofaciens* is able to ferment a wide range of carbohydrates and to produce H_2_, ethanol, lactate and short-chain fatty acids [38]. Several studies demonstrate that *C. aerofaciens* is able to modulate the pathogenicity of enteric pathogens and to ameliorate symptoms in patients affected by irritable bowel syndrome [39,40,41]. In addition, a recent study reported a diminished abundance of *C. aerofaciens* in IBD patients [42]. Therefore, the presence of *C. aerofaciens* during clinical remission time-points and its absence or decrease during clinical relapse may suggest that this species can play a positive role in UC patients.

Hierarchical cluster analyses showed two distinct microbial ecologies of recipients, which probably reflected their different clinical signs and symptoms. Interestingly, P2’ samples at T_2_ clustered together with P1’ samples, thus indicating that, at this time point, gut microbial architecture was closer to a microbiota profile belonging to attenuated symptomatic conditions.

The group 1 cluster was characterized by the significant presence of *P. copri*, together with *Enterococcus* spp. and *Lactobacillus* spp. *P. copri* has been associated with chronic immune-mediated inflammatory diseases, such as rheumatoid arthritis in humans and with exacerbation of colitis in mice models [43,44]. This classification of ecological clusters, based on microbiota composition, did not reflect patients’ clinical condition before and after FMT. However, comparison of the RE and AD sample groups revealed that *Proteus*, *B. producta* and *R. gnavus* were exclusively in AD and absent from RE, controls and donors profiles. A previous study reported that *Proteus* was strongly associated with IBD. This microorganism could exert its pathogenicity by releasing bacterial products in proximity to the apical epithelial surface, triggering an inflammatory response [45]. Authors argued that *Proteus* could elicit colitis in concert with other unidentified members of microbiota in IBD patients. Interestingly, *Proteus* was present only in P2’ samples and its presence correlated with patient’s clinical conditions. Indeed, it was highly abundant at T_0_, then disappeared when clinical conditions were stable (T_1_ and T_2_) and finally returned to its initial relative abundance at T_3_, when the clinical condition worsened. However, the relationship between clinical course and bacterial abundancies was followed also by *B. producta* and *R. gnavus* in P2. In P1, among the three statistically significant species, only *B. producta* seemed to correlate with patient’s clinical condition. Indeed, *Proteus* was completely absent, supporting the hypothesis that it may be associated only with more moderate clinical conditions of UC. *R. gnavus* decreased after FMT, increased at T_2_, showing a trend completely independent from the patient’s symptoms. Moreover, *Parabacteroides*, *Mogibacteriaceae*, *B. eggerthi*, *B. plebeius*, *R. bromii*, and *B. ovatus* were incremented in RE group. Notably, all the species found in the RE group were shared by donors and control microbiota. Our results confirm the conclusions of Kellermayer that the presence in donors of *Bacteroides*, *Eubacterium* and *Ruminococcus* could be considered predictive of therapeutic success of FMT [37]. Some questions remain open and require further investigations to evaluate multiple FMT efficacy for the moderate forms, to decide the right timing of various administrations and to assess intra-familiar or unrelated donors as the best choice for pediatric FMTs in IBD.

## 5. Conclusions

Despite being focused only on two cases, our results seem to suggest that FMT could be considered effective for mild UC patients in need of therapeutic step-up. Moreover, single administration of FMT could be suitable for UC mild forms, and multiple FMTs could be appropriated for amelioration or remission of UC moderate forms. Finally, our results led us to infer that the increment of some microorganisms correlated to patient’s remission, and the decrease of other potentially pathogens, could contribute to the therapeutic success of FMT in pediatric UC.

## Figures and Tables

**Figure 1 microorganisms-08-01486-f001:**
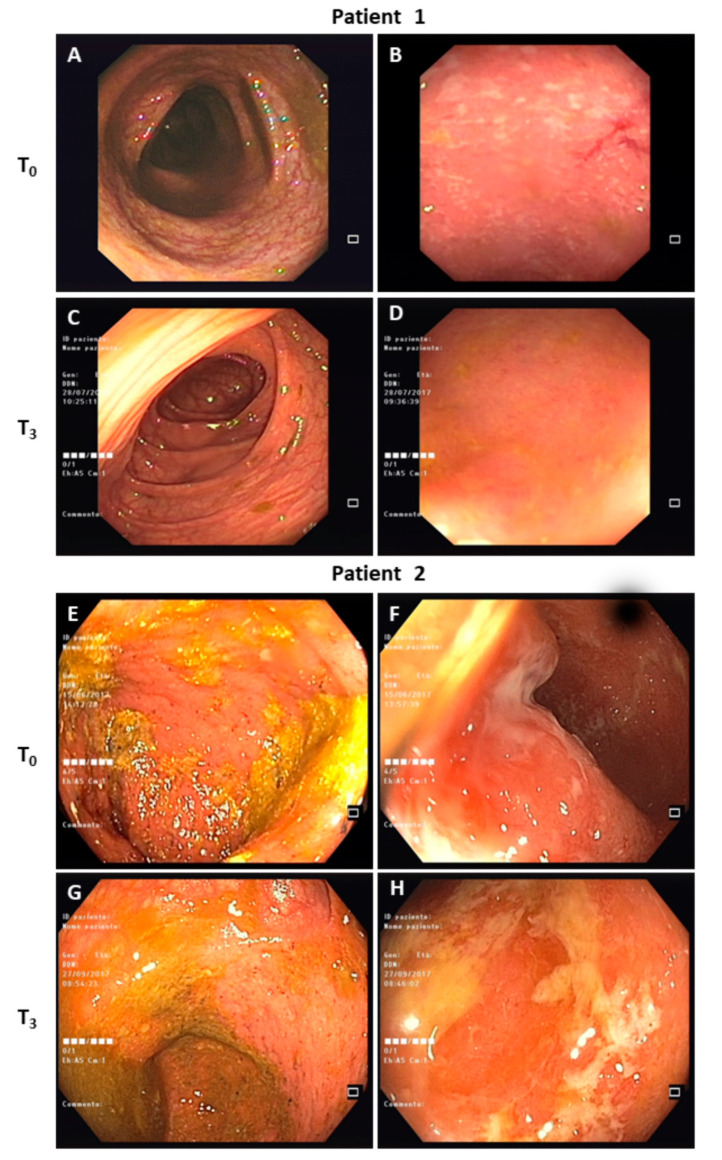
**Colonoscopy images of Patients 1 and 2 at the time of FMT (T_0_) and after 12 weeks (T_3_)**. Patient 1. Endoscopic appearance at T_0_: ascending colon, Mayo score 0 (**A**), and rectum, Mayo score 2 (**B**). Endoscopic appearance at T_3_: ascending colon, Mayo score 0 (**C**), and rectum, Mayo score 1 (**D**). Patient 2. Endoscopic appearance at T_0_: ascending colon, Mayo score 1 (**E**), and rectum, Mayo score 2 (**F**). Endoscopic appearance at T_3_: ascending colon, Mayo score 1 (**G**), and rectum, Mayo score 2 (**H**). Endoscopic Mayo score. Score 0: normal mucosa, pale pink, smooth, and glistening, with evident submucosal blood vessels; score 1: erythema, decreased vascular pattern, mild friability; score 2: marked erythema, absent vascular pattern, friability, erosions (fibrinous exudate appearing as whitish material); score 3: spontaneous bleeding and ulcerations.

**Figure 2 microorganisms-08-01486-f002:**
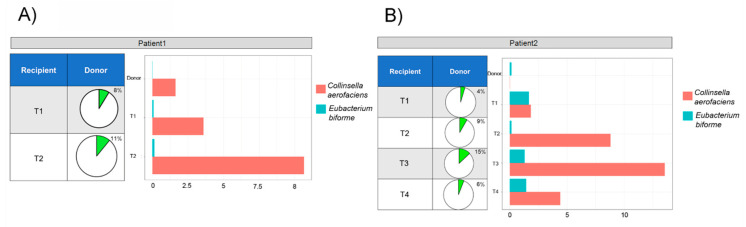
Donor’s OTUs inherited from patients. The percentage and relative abundance of donor species within P1 (**A**) and P2 (**B**) microbiota acquired through FMT. Pie charts indicate sum of percentage of *Collinsella aerofaciens* plus *Eubacterium biforme* within the whole microbiota profile, while the bar charts indicate their relative abundance.

**Figure 3 microorganisms-08-01486-f003:**
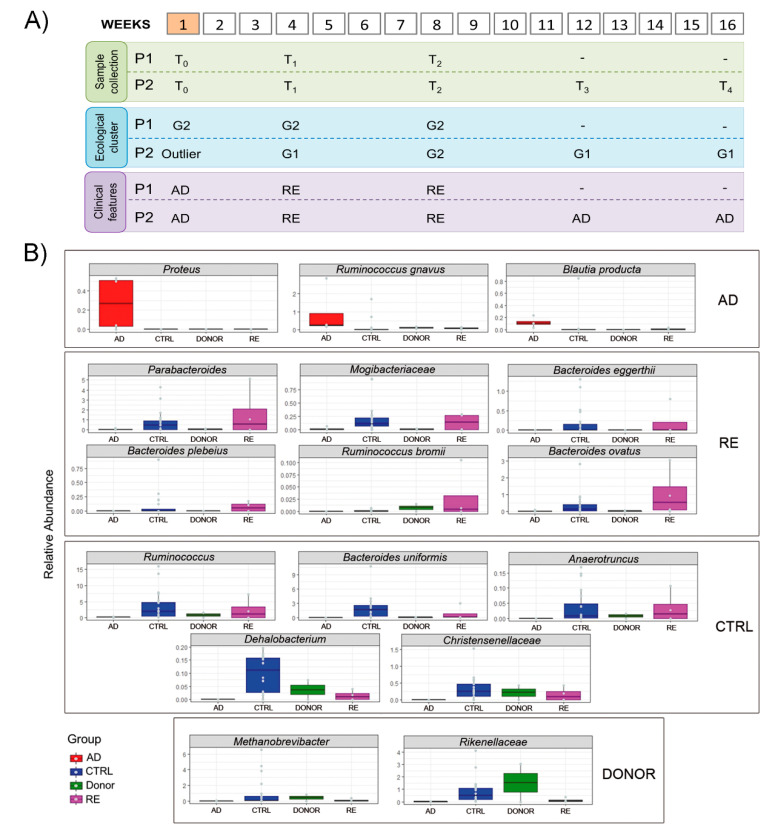
Experimental timeline and sample clusterization, based on ecological and clinical evaluations. **Panel A.** Samples collection: for both patients, sample collection timeline is expressed in weeks after FMT (orange box). Ecological cluster: hierarchical cluster analysis, based on microbiota profile, with samples divided into two groups (G1 and G2). Clinical features: samples were classified basing on patient clinical features (RE, indicates remission status and AD, active disease status). **Panel B.** Bar charts compare the relative abundance of 16 bacterial species, obtained by DESeq2 analysis, for the patients’ samples collected during RE or AD and from CTRLs and donors to describe which species were associated to patient’s clinical status.

**Table 1 microorganisms-08-01486-t001:** Clinical features of P1 and P2 patients. Calprotectin and PUCAI pre-FMT ranges refer to measurement performed in the three months before transplantation, while Calprotectin and PUCAI post-FMT values refer to T_0_–T_4_ time points.

Features	P1	P2
**UC extension**	Left-sided colitis	Pancolitis
**Stool Donor**	Single	Single
**Patient Age (years)**	16	15
**Donor Age (years)**	55	48
**Family Relationship**	Father	Father
**Stool processing**	Fresh	Fresh
**Route of delivery**	Colonoscopy	Colonoscopy
**FMT regimen**	One infusion	One infusion
**Patient condition before FMT (screening)**	Recurrent episodes of bloody diarrhea and abdominal pain, needing multiple cycles of steroids/antibiotics	Recurrent episodes of bloody diarrhea and abdominal pain, needing multiple cycles of steroids/antibiotics
**Maintenance therapy before FMT**	Mesalazine	Mesalazine + Azathioprine
**Antibiotic treatment before FMT (4 weeks)**	No	No
**Patient condition at the time of FMT (T_0_)**	Diarrhea without blood	Semi-formed stool with occasional blood
**PUCAI pre-FMT**	15–20	3–35
**MAYO Score pre-FMT**	Mayo score: 0 in the ascending-transverse-descending colon, 2 in the sigmoid-rectum	Mayo score: 1 in the ascending-transverse colon, 2 in the descending colon-sigmoid-rectum
**Calprotectin pre-FMT**	81–187 g/gr	>500 g/gr
**PUCAI post-FMT**	0	5–25
**MAYO post-FMT**	Mayo score: 0 in the ascending-transverse-descending colon, 1 in the sigmoid-rectum	Mayo score: 1 in the ascending-transverse colon, 2 in the descending colon-sigmoid-rectum
**Calprotectin post-FMT**	61–248 g/gr	138–237 g/gr
**Antibiotic treatment post FMT**	No	Yes (at 12 weeks, Metronidazole)
**Clinical remission post FMT**	Yes	No (at 6 weeks)
**Maintenance therapy post-FMT**	Mesalazine	Mesalazine + Azathioprine
**Patient condition at T_1_ (4 weeks)**	Absence of symptoms	Absence of symptoms
**Patient condition at T_2_ (8 weeks)**	Absence of symptoms	Absence of symptoms
**Patient condition at T_3_ (12 weeks)**	Absence of symptoms	Relapse of bloody diarrhea (brief cycle of Metronidazole)
**Patient condition at T_4_ (16 weeks)**	Absence of symptoms	Semi-formed stool without blood

As a minor deviation to the protocol, stool sample was missed for P1 at 12 weeks (T_3_), due to noncompliance in sample collection, while the adjunctive sample was collected for patient 2 at 16 weeks (T_4_), considering the option of a booster FMT.

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
