# Peer review of "Fecal Microbiota Transplant in Two Ulcerative Colitis Pediatric Cases: Gut Microbiota and Clinical Course Correlations"

_microorganisms, 2020, doi:10.3390/microorganisms8101486_

Round 1
Reviewer 1 Report
The manuscript described an innovative therapeutic method for treatment of ulcerative colitis (UC) using Fecal microbiota transplantation (FMT). The study is well organized and presented in a time-manner systematically using different statistical methods. The reviewer highly recommend the acceptance of the manuscript after the following minor revisions:
- The names of microorganisms in Figure 2 should be enlarged. The word is too small to see clearly in print situation.
- Figure1 : P1 T3C compared with Figure T0 A, the blood vascular vessels disappeared, does that mean the inflammation reduced? Also P1 T3D compared with T0B, the white cloudy substance reduced, does that mean UC condition ameliorate? Can the author further explain it in text or in figure to facilitate the change of P1 in addition to Mayo score?
- Also, Figure 3B, the names of microorganisms is too small to read. It should be enlarged. In addition, can authors arrange the Figure 3B in a way that is more easy to understand? For example, the microorganisms increased in RE compared with CTRL put in one row while the microorganisms decreased RE compared with CTRL put in another row?
Author Response
The manuscript described an innovative therapeutic method for treatment of ulcerative colitis (UC) using Fecal microbiota transplantation (FMT). The study is well organized and presented in a time-manner systematically using different statistical methods. The reviewer highly recommend the acceptance of the manuscript after the following minor revisions:
Author’s reply:
We really thank you to appreciate our work.
We also thank you for having highlighted some inaccuracies in the text and having provided us with suggestions.
Following your suggestions, we have modified the manuscript by “Track change modality” to identify easily the changes.
The names of microorganisms in Figure 2 should be enlarged. The word is too small to see clearly in print situation.
Author’s reply:
Done, please consider the new version of the manuscript in which the figures have been amended, including Figure 1.
Figure 1 : P1 T3C compared with Figure T0 A, the blood vascular vessels disappeared, does that mean the inflammation reduced? Also P1 T3D compared with T0B, the white cloudy substance reduced, does that mean UC condition ameliorate? Can the author further explain it in text or in figure to facilitate the change of P1 in addition to Mayo score?
Author’s reply:
Evidence of blood vessels is a sign of mucosal integrity and it is present in both images T0A and T3C (even if clearer in T3C than in T0A), both showing normal colonic mucosa. Conversely, decreased/absent vascular pattern is a sign of inflammation, evident in both images T0B and T3D, showing inflamed rectum. Another typical sign of inflammation is the whitish material that indicates fibrin exudate/erosions. This characteristic is present in T0B but not in T3D, as a sign of inflammation reduction.
The figure legend has been implemented with a written description of endoscopic Mayo score.
Also, Figure 3B, the names of microorganisms is too small to read. It should be enlarged. In addition, can authors arrange the Figure 3B in a way that is more easy to understand? For example, the microorganisms increased in RE compared with CTRL put in one row while the microorganisms decreased RE compared with CTRL put in another row?
Author’s reply:
Done, please consider the new version of the manuscript in which the figures have been amended, including Figure 3.
Reviewer 2 Report
Dr. Quagliariello et al. submitted their paper entitled "Fecal Microbiota Transplant in two Ulcerative Colitis pediatric cases: gut microbiota and clinical course correlations.
Fecal Microbiota Transplantation (FMT) was developed mainly to cure patients that suffered from recurrent infection with Clostridium difficile. FMT shows high effectivity. Knowledge dealing with microbiota and its influence on the health of the host exponentially increased during the last decade. Inflammatory bowel diseases as ulcerative colitis and Crohn's disease are related to the intestinal microbiota. It is possible to expect that FMT will show a positive effect on IBD. New knowledge that helps to develop and standardize FMT for the IBD treatment is required. It is the reason why I evaluate the submitted brief report manuscript interesting and highly actual.
The authors described mild and moderate ulcerative colitis in children. It seems that the used treatment can be useful in mild but not in moderate colitis. The improvement based probably on an altered treatment protocol of FMT for more severe illness is needed.
I have only several notices and recommendations:
L2-4: I don't understand using capital letters in the title. E.g., pediatric cases versus Pediatric Cases. It would be unified through the whole title.
L61: "Recently". FMT to cure recurrent infections with C. difficile has been used at least one decade. It is not a method that was developed recently. Please, rephrase the sentence.
L93: age 18-70 years. Microbiota of elders is distinct from younger people. I think that criteria, at least for children and youth, should be limited by the lower age than 70 years.
L104: I am not sure if the cecum is the best place to cure the colonic disease. Will you justify it as your response to me, please?
L131-135: This text should not be included in the statistical analysis.
Figure 2: It is necessary to increase fonts for the description of columns (orange and green).
Figure 3: The figure is described by tiny letters. Please, increase the font. It should be comparable with the font of the figure legend.
Author Response
Dr. Quagliariello et al. submitted their paper entitled "Fecal Microbiota Transplant in two Ulcerative Colitis pediatric cases: gut microbiota and clinical course correlations.
Fecal Microbiota Transplantation (FMT) was developed mainly to cure patients that suffered from recurrent infection with Clostridium difficile. FMT shows high effectivity. Knowledge dealing with microbiota and its influence on the health of the host exponentially increased during the last decade. Inflammatory bowel diseases as ulcerative colitis and Crohn's disease are related to the intestinal microbiota. It is possible to expect that FMT will show a positive effect on IBD. New knowledge that helps to develop and standardize FMT for the IBD treatment is required. It is the reason why I evaluate the submitted brief report manuscript interesting and highly actual.
The authors described mild and moderate ulcerative colitis in children. It seems that the used treatment can be useful in mild but not in moderate colitis. The improvement based probably on an altered treatment protocol of FMT for more severe illness is needed.
Author’s reply:
We really thank you to appreciate our work.
We also thank you for having highlighted some inaccuracies in the text and having provided us with suggestions.
Following your suggestions, we have modified the manuscript by “Track change modality” to identify easily the changes.
I have only several notices and recommendations:
L2-4: I don't understand using capital letters in the title. E.g., pediatric cases versus Pediatric Cases. It would be unified through the whole title.
Author’s reply:
Done, we removed the capital letters in the title.
L61: "Recently". FMT to cure recurrent infections with C. difficile has been used at least one decade. It is not a method that was developed recently. Please, rephrase the sentence.
Author’s reply:
Done, please consider the new sentence.
L93: age 18-70 years. Microbiota of elders is distinct from younger people. I think that criteria, at least for children and youth, should be limited by the lower age than 70 years.
Author’s reply:
Thanks for the comment. The donor age range reported in the manuscript is a typo. The FMT Standard Operating Procedures adopted in our hospital are aligned to guidelines for FMT (Cammarota et al., Gut 2019 Dec;68(12):2111-2121. doi: 10.1136/gutjnl-2019-319548; Cammarota et al., Gut . 2017 Apr;66(4):569-580. doi: 10.1136/gutjnl-2016-313017. Epub 2017 Jan 13) in which was reported that individuals aged <60 years should be preferred as FMT donors. Please consider the new version of the manuscript.
L104: I am not sure if the cecum is the best place to cure the colonic disease. Will you justify it as your response to me, please?
Author’s reply:
Cecum, as known, is the first segment of the colon. Since peristaltic movements are directed from cecum to the next segments, this is the best place to release the fecal suspension, in order to colonize all of them. Even if the inflammation is only in the last segments, the entire colonic microbiota can be therefore restored.
L131-135: This text should not be included in the statistical analysis.
Author’s reply:
Done, we moved the indicated text in previous paragraph.
Figure 2: It is necessary to increase fonts for the description of columns (orange and green).
Author’s reply:
Done, please consider the new version of the manuscript in which the figures have been amended, including Figure 2.
Figure 3: The figure is described by tiny letters. Please, increase the font. It should be comparable with the font of the figure legend.
Author’s reply:
Done, please consider the new version of the manuscript in which the figures have been amended, including Figure 3.